# Impact of COVID-19 Outbreak on Influenza and Pneumococcal Vaccination Uptake: A Multi-Center Retrospective Study

**DOI:** 10.3390/vaccines11050986

**Published:** 2023-05-15

**Authors:** Chieh Lan, Yi-Chun Chen, Ye-In Chang, Po-Chun Chuang

**Affiliations:** 1Department of Family Medicine, Kaohsiung Chang Gung Memorial Hospital, Kaohsiung 83301, Taiwan; lanchieh33@cgmh.org.tw; 2Division of Infectious Diseases, Department of Internal Medicine, Kaohsiung Chang Gung Memorial Hospital, Kaohsiung 83301, Taiwan; sonice83@cgmh.org.tw; 3Department of Computer Science and Engineering, National Sun Yat-sen University, Kaohsiung 80424, Taiwan; changyi@mail.cse.nsysu.edu.tw; 4Department of Emergency Medicine, Kaohsiung Chang Gung Memorial Hospital, Kaohsiung 83301, Taiwan

**Keywords:** COVID-19, pneumococcal vaccine, influenza vaccine

## Abstract

During the coronavirus disease 2019 (COVID-19) pandemic, global vaccination efforts declined due to the burden on health systems and community resistance to epidemic control measures. Influenza and pneumococcal vaccines have been recommended for vulnerable populations to prevent severe pneumonia. We investigated community response towards influenza and pneumococcal vaccines (pneumococcal conjugate vaccine and pneumococcal polysaccharide vaccine) after the COVID-19 outbreak in Taiwan. We retrospectively included adults who visited Chang Gung Memorial Hospital (CGMH) institutions for influenza or pneumococcal vaccination from January 2018 to December 2021. The first case of COVID-19 in Taiwan was detected in January 2020; therefore, in this study, hospitalized cases from January 2018 to December 2019 were defined as “before COVID-19 outbreak,” and hospitalized cases from January 2020 to December 2021 were defined as “after COVID-19 outbreak”. A total of 105,386 adults were enrolled in the study. An increase in influenza vaccination (*n* = 33,139 vs. *n* = 62,634) and pneumococcal vaccination (*n* = 3035 vs. *n* = 4260) were observed after the COVID-19 outbreak. In addition, there was an increased willingness to receive both influenza and pneumococcal vaccinations among women, adults without underlying disease and younger adults. The COVID-19 pandemic may have increased awareness of the importance of vaccination in Taiwan.

## 1. Introduction

Routine immunization of adults has received increased attention in recent years due to the recognition of the substantial burden of vaccine-preventable diseases in this age group. In the US, routine annual influenza vaccination is recommended for all persons aged ≥6 months who do not have contraindications. Pneumococcal polysaccharide vaccines are recommended for elderly patients and those with specific medical or other conditions. This should be followed by 1 year of the recommended dose of pneumococcal conjugate vaccine (PCV) for otherwise healthy adults aged 65 years or older. A single dose of PCV is recommended for adults aged 19 years and older with high-risk conditions such as functional or anatomic asplenia, immunosuppression (including hematologic malignancy, generalized malignancy, and immunosuppressive medications), renal disease (chronic renal failure or nephrotic syndrome), cochlear implants, or cerebrospinal fluid leakage.

Some studies reported that during the coronavirus disease 2019 (COVID-19) pandemic, routine immunization activities were at risk of disruption due to burden on the health system, reduced need for vaccination owing to physical distancing requirements, and community resistance [1]. In the United States, several studies have reported a decline in routine vaccination doses (non-influenza) due to the COVID-19 pandemic [2,3]. Additionally, the fear of contracting severe acute respiratory syndrome coronavirus 2 (SARS-CoV-2) in a medical setting may discourage families from seeking medical care, including annual influenza vaccinations. On the other hand, there has been considerable concern about the safety and efficacy of the COVID-19 vaccine based on individual surveys [4,5]. People may choose vaccines that protect against other infectious respiratory diseases instead of COVID-19 vaccination for compensation. In a UK-wide observational study, COVID-19 resulted in substantial uptake of influenza vaccination among previously eligible yet not vaccinated individuals [6]. According to some researchers, the incidence of COVID-19 in the elderly (>65 years old) may be reduced by influenza vaccination [7]. Furthermore, educating the public about recent evidence suggesting that the influenza and SARS-CoV-2 co-infection doubles mortality compared to the infection with SARS-CoV-2 alone has raised public awareness of influenza vaccines in the United Kingdom [8]. Although there is still no solid evidence of how influenza vaccines may or may not relate to COVID-19 infection, to our knowledge, we believe these multifaceted discussions represent the public’s attention to seeking varied solutions for COVID-19.

A common complication of COVID-19 is bacterial co-infection [9]. Reports have suggested that SARS-CoV-2 may enhance bacterial colonization and attachment to host tissues, and that concurrent infections may cause irreversible tissue damage and exacerbate pathology [10]. Furthermore, the prevalence of COVID-19-associated co-infections and secondary infections is up to 45% [11], and secondary bacterial infections are responsible for half of all deaths [12]. Additionally, bacterial co-infection increases the risk of death in COVID-19 patients by 5.82 times compared to those without co-infection [13]. A recent Mayo clinic study [14] reported that pneumococcal 13-valent conjugate vaccine (PCV13)-vaccinated adults acquired certain pneumococcal strains and were 35% less likely to be infected with COVID-19 than non-vaccinated adults.

This study investigated whether the COVID-19 pandemic has changed the public behavior towards influenza vaccine and pneumococcal vaccine in Taiwan. In this research, we compared the demographics and underlying health conditions of individuals who received influenza and pneumococcal vaccinations between 2018–2019 and 2020–2021.

## 2. Materials and Methods

### 2.1. Participants

A total of 105,386 adults (age ≥ 17 years-old), who visited the Chang Gung Memorial Hospital (CGHM) institutions and received an influenza or pneumococcal vaccine between January 2018 and December 2021, were enrolled in the study.

In Taiwan, government-funded annual influenza vaccines are provided to individuals aged >50 years, healthcare workers, and people with risk factors for severe influenza development. Government-funded pneumococcal vaccines include pneumococcal polysaccharide vaccine (PPSV23, Pneumovax^®^ NP Merck & Co., Inc., Kenilworth, NJ, USA) and are provided to senior citizens, as defined by the local government. Adults may choose PCV13 (Prevenar13^®^ Pfizer Inc., New York, NY, USA) at their own expense. Influenza vaccine included Vaxigrip Tetra^®^ (Sanofi Pasteur, Val de reuil Cedex, France), Fluarix Tetra^®^ (GlaxoSmithKline Biologicals, Dresden, Germany), AdimFlu-S (QIS)^®^ (Adimmune Inc., Taichung, Taiwan) and FLUCELVAX QUAD^®^ (CSL Behring GmBH, Marburg, Germany). These vaccines were provided randomly to public in Taiwan according to the purchase amount by government since October every year.

### 2.2. Data Collection

The Chang Gung Research Database (CGRD) [15] contains information on patients who received influenza or pneumococcal vaccinations in medical institutions of the CGMH system. We collected the data retrospectively by filtering cases with the International Statistical Classification of Diseases and Related Health Problems 10th Revision (ICD-10) and prescription numbers (including influenza vaccine, PCV13 and PPSV23) in CGRD. All the information was de-identified and aggregated for analysis. As a result, we did not obtain specific consent for publication from the patients who participated in the study. This research included four medical institutes located in southern and northern Taiwan (Kaohsiung, Chiayi, Linkou, and Keelung branches).

### 2.3. Outcomes Measure

Individuals with multiple visits to CGMH institutions were counted as multiple cases. For example, a person who visited a CGMH institution and received an influenza vaccination in 2018 and 2019 was counted as two cases. In Taiwan, patients who came for vaccinations were required to have their health identification cards, and all cases had to be registered while receiving any type of vaccines. As such, healthcare providers could check the timing of vaccines and prevent administering repeat shots of the same type of vaccine within the wrong schedule. This method helps avoid administering two influenza vaccines in a year. Moreover, according to Taiwan Centers of Disease Control (CDC), government-funded PPSV23 has been provided to individuals >75 years old once in a lifetime since 2007, and those who received PPSV23 first can get PCV13 after at least a year, and vice versa [16]. Hence, in our outcomes, we might find individuals who came to our institution twice in different years. COVID-19 was first detected worldwide in December 2019, and the first reported case in Taiwan was detected in January 2020 [17,18,19]. Therefore, in this study, hospital cases from January 2018 to December 2019 were defined as “before COVID-19 outbreak,” and hospital cases from January 2020 to December 2021 were defined as “after COVID-19 outbreak.”

### 2.4. Data Analysis

Age, as a continuous variable, was presented as the mean with standard deviation. Categorical variables were presented as numbers and percentages. The *t*-test was used to analyze continuous variables. The chi-square test was used to analyze categorical variables. Results were considered statistically significant for a two-tailed test if the *p*-value was <0.05. All statistical analyses were performed using SPSS for Windows (version 22.0, 2013, IBM Corp., Armonk, NY, USA).

## 3. Results

### 3.1. Clinic Population and Duration

Outpatient visits at the Keelung, Linkou, Chiayi, and Kaohsiung branches of the Chang Gung institution were analyzed. The total number of outpatient visits in 2018, 2019, 2020, and 2021 were 1,590,976, 1,894,144, 1,731,762, and 1,759,111, respectively. This study defines January 2020 as the time of the COVID-19 outbreak. A total of 105,386 adults (>17 years old) who received the influenza vaccine or pneumococcal vaccine (PCV13 or PPSV23) were enrolled in this retrospective study from 1 January 2018 to 31 December 2021; 95,773 and 9613 cases were immunized with the influenza and pneumococcal vaccines (PCV13 or PPSV23), respectively (Figure 1). Overall, the number of people vaccinated against influenza was 33,139 (0.95%) and 62,634 (1.79%) before and after the outbreak (*p* < 0.001). The number of people who received pneumococcal vaccines was 3035 (0.09%) before the outbreak compared with 4260 (0.12%) after the outbreak (*p* < 0.001).

### 3.2. Influenza Vaccination Cases before and after COVID-19

An increasing number of cases was observed after COVID-19 (*n* = 33,139 vs. *n* = 62,634, in 2018–2019 and 2020–2021, respectively). We found significant differences between age, gender, location, and all underlying diseases (*p* < 0.001) (Table 1).

The location distribution shifted to more cases in the Keelung, Linkou, and Chiayi institutions (Table 1). Linkou and Kaohsiung branches are medical centers, which contain 3700 and 2640 beds, respectively, while Keelung and Chiayi branches are regional hospitals with fewer than 2000 beds. We noticed that all cases increased in different locations after the COVID-19 outbreak, but the two medical centers still constituted the majority. In addition, patients without a history of underlying diseases showed an increased willingness to receive influenza vaccination. Since the Taiwan Centers for Disease Control announced influenza vaccination every October, we observed that the number of cases increased in October after the COVID-19 outbreak (Figure 2a). In Figure 3, we compared case numbers in different years and different seasons. Data shows apparent surge of influenza vaccine administration in 2021 (Figure 3a). Also, based on data from the Taiwan Centers for Disease Control, we found that cases of severe complicated influenza decreased significantly after COVID-19 outbreak in 2020 and 2021 (Figure 3b).

### 3.3. Pneumococcal Vaccination Cases before and after COVID-19

An increasing number of cases was observed after the outbreak of COVID-19 (*n* = 3035 vs. *n* = 4260, in 2018–2019 and 2020–2021 respectively). There were significant differences between patients who received a pneumococcal vaccine (PCV13 or PPSV23) after COVID-19 and those who did not (*p* < 0.001). In addition, there was a significant difference (*p* < 0.001) between patients who received PCV13 vaccination more frequently than PPSV23 vaccination after the outbreak. Patients vaccinated with the pneumococcal vaccine after the COVID-19 outbreak had a lower proportion of hypertension than before the outbreak (*p* < 0.001). Furthermore, patients without underlying diseases demonstrated greater interest in receiving pneumococcal vaccination (Table 2). For the monthly comparison, we analyzed elevated cases, except in December, after the COVID-19 outbreak (Figure 3b).

### 3.4. Influenza and Pneumococcal Vaccination around the COVID-19 Outbreak Point in Taiwan

Between 2018 and 2021, the number of influenza vaccinations was the highest every October, except for 2019. We observed low numbers of influenza vaccine in October 2019, but a dramatic increase from December 2019 to January 2020. For pneumococcal vaccination, we observed distinctly increased number of vaccinations between December 2019 and January 2020, and between October 2020 and June 2021 (Figure 2). This pattern suggests that the COVID-19 outbreak may have influenced the public’s interest in receiving both influenza and pneumococcal vaccinations in Taiwan, with an increased number of vaccinations administered around the outbreak point and throughout the subsequent months.

## 4. Discussion

### 4.1. Increased Willingness for Pneumococcal Vaccination after the COVID-19 Pandemic

In our study, we found elevated willingness for pneumococcal vaccination after the COVID-19 outbreak. This might be due to the public’s awareness of severe pneumonia. The WHO had already claimed that the most common diagnosis for COVID-19 is severe pneumonia due to acute respiratory infection [20]. Furthermore, bacterial coinfection or secondary bacterial infections are common complications of respiratory viral disease [21]. Hence, the importance of patients be immunized to reduce the risk of preventable coinfections with other etiology were emphasized. These perspectives brought people tend to prevent not only COVID-19 but also other pathogens that might cause severe pneumonia, and that might be one of the reasons why Taiwanese paid more attention to pneumococcal vaccination.

Moreover, several studies conducted in children and adults showed that pneumococci were associated with virus-associated respiratory diseases, including the human coronavirus, suggesting their role as a causative agent [22,23,24]. Another study in France even claimed that COVID-19 pandemic is an opportunity to increase in influenza and pneumococcal vaccine coverage in the at-risk population [25]. We believed that even though there was no certain findings of pneumococcal vaccine and better COVID-19 outcomes, the role of public behaviors over COVID-19 prevention and pneumococcal vaccine is worth exploring.

### 4.2. Increased Desire for Influenza Vaccination after the COVID-19 Pandemic

Since COVID-19 and influenza are both respiratory infectious diseases caused by enveloped RNA viruses that have similar transmission routes and clinical characteristics [26], there has been a growing interest in the relationship between influenza immunity and SARS-CoV-2 infection among researchers. In their cohort study, Maor et al. reported an increased willingness to receive an influenza vaccine between 2020 and 2021 compared to the 2019–2020 season, with an increase of 22.6% in people aged less than 65 years and an increase of 7.3% in people aged 65 years and older in the Jewish Israeli population [27]. This report found similar results to our study in that more adults tended to accept the influenza vaccine after 2020.

We surmised that the public might be more interested in getting influenza vaccine because of the vivid discussion over severe pneumonia under COVID-19 pandemic, which is the same reason we mentioned previously for the influenza vaccine as well. Meanwhile, one study found that patients who were influenza-vaccinated required fewer hospitalizations, less mechanical ventilation, and shorter hospital stays in the US [28]. Also, several recent studies have suggested that prior vaccination against pathogens, such as tuberculosis and influenza, may provide protection against COVID-19 [7,29,30,31,32]. As a result, these suggestions were publicized via media and may encourage the public to receive influenza vaccination during the COVID-19 pandemic. However, we have noticed several studies tried to find relationship between higher influenza vaccination rates and better COVID-19 outcomes, but there was no concrete explanation and proof [33,34,35]. Meanwhile, according to the US CDC, they advised interim guidance for influenza vaccination during the COVID-19 pandemic, which indicates reducing the overall burden of respiratory illness is important to vulnerable populations, and healthcare personnel should use every opportunity during the influenza seasons to administer influenza vaccines to all eligible persons [36]. Also, Taiwan CDC had suggested that the influenza vaccine and COVID-19 vaccination can be taken at the same time [37]. Hence, under the health care providers’ promote and the convenience to administering both vaccines simultaneously, that might be the reason we found an obvious surge of cases coming for influenza vaccine after 2020 in Taiwan.

### 4.3. Underlying Diseases in Women after the COVID-19 Pandemic

An increasing percentage of females are getting both the pneumococcal and influenza vaccine after the COVID-19 outbreak (Table 1 and Table 2). A previous study claimed that males are more likely to display vaccine hesitancy in general [38]. Another study in Japan that searched for factors related to influenza vaccine inoculation and non-inoculation behavior after the COVID-19 outbreak found significantly lower odds of vaccination in males compared to females [39]. Another study in Japan revealed that women aged 20–64 years showed a significantly higher percentage of influenza vaccine inoculation after COVID-19 [40]. Nevertheless, in contrast to our results, previous studies have discussed vaccine hesitancy, which revealed that females were more hesitant toward influenza [41] and COVID-19 vaccines [42]. There was little speculation or proof of whether females were more likely to be vaccinated, but we believe that gender might play a complicated role in the willingness to administer vaccines.

Adults without underlying diseases have shown an elevated interest in getting vaccinated after the COVID-19 outbreak for both the pneumococcal and influenza vaccines. In Japan, Komada et al. found that those aged 20–64 years showed a significantly higher percentage of influenza vaccine inoculation, including women and those who visited the hospital for lifestyle-related illnesses (including diabetes, angina, myocardial infarction, asthma, hypertension, stroke, gout, hepatitis, cirrhosis, hypercholesterolemia, hypertriglyceridemia, hyperuricemia, and stomach/duodenal illnesses) and orthopedic illnesses than those without lifestyle-related illnesses that visited the hospital [39]. Another cross-sectional survey study in Spain revealed that the COVID-19 pandemic has prompted doctors and nurses to engage patients with vaccines, particularly in terms of prioritizing vaccination as a preventive measure, increasing public awareness of vaccination, and providing additional vaccination training to general practitioners [40]. In this case, we assumed that increasing awareness among healthcare professionals of pneumococcal and influenza vaccines might lead to healthy adults (without underlying disease) being vaccinated at higher rates as well. In addition, several studies have indicated that adult participants’ willingness to be vaccinated has a strong correlation when recommended by their healthcare providers [43,44,45,46,47]. Furthermore, one recent study conducted in Taiwan revealed the following factors may enhance vaccination intentions: male, older, having an open personality, having better physical health quality of life, trusting the government, and the author also mentioned that when the pandemic intensified, Taiwanese’ vaccination intentions increased significantly. [48] In our knowledge, vaccination motivations can be affected by not only genders, but also psychological perspectives, health-related behaviors, political attitudes, and COVID-related risk factors, including the location of people, quarantine experiences, and the number of new cases confirmed on a daily basis. In the future, governments can design targeted strategies to increase vaccination rates based on the findings of the present study, which could shed light on individuals’ vaccination attitudes.

### 4.4. Increasing Rate of Vaccination in Taiwan in Contrast to the Global Vaccination Disruption

During the COVID-19 pandemic, medical services and vaccinations were disrupted in more than 90% of countries [49,50]. For example, a reduction in measles-mumps-rubella vaccination was noted in England in early 2020 [51], a decline in the administration of non-influenza childhood vaccine doses in Michigan in 2020 [52], and a decrease in routine vaccinations was observed in Pakistan during the pandemic [53]. In fact, when COVID-19 was raging, vaccination disruptions were common and significant. This may be due to several factors, including disruptions in medical services or vaccination practices, lockdowns, school closures, stay-at-home policies, insufficient supplies of personal protective equipment, and a global lack of medical staff and healthcare providers.

In contrast, COVID-19 was well-controlled in Taiwan before May 2021; hence, an earlier study on Taiwanese participants found that vaccination was unnecessary or not urgent [54]. However, after the COVID-19 outbreak in May 2021, Taiwanese people rushed to receive the COVID-19 vaccine. As public anxiety increased during the epidemic, people looked for surrogate vaccinations against COVID-19 because of the shortage of COVID-19 vaccines and the detected risk of infection. Meanwhile, an article published in mid-March 2021 reported a lower risk of COVID-19 in individuals vaccinated with a pneumococcal conjugate vaccine (PCV13) [55]. As a result of the media coverage from this article, Taiwanese medical authorities endorsed PCV13 vaccination as beneficial. Therefore, Taiwanese people obtained PCV13 vaccines to compensate for the COVID-19 vaccine shortage [54], and we assumed that the influenza vaccine and PPSV23 received similar attention, although no study has proved this so far.

Besides, we found the percentage of people who came for influenza vaccine and pneumococcal vaccine among all outpatient visits were significantly elevated as well. The number of outpatient consultations will be affected by many factors, such as the infection control of the hospital, and the public’s fear of possible contact with COVID-infected people when seeking medical treatment, etc. This result also verified our perspective that even though the COVID-19 pandemic changed public behaviors of visiting hospitals, we still noticed an increasing interest of immunization after the COVID-19 outbreak in Taiwan proportionally.

Furthermore, Taiwan’s health insurance program is distinctive in that it covers a broad range of residents, resulting in good accessibility, low cost, comprehensive coverage, and short waiting times [56]. Before the COVID-19 pandemic, Taiwan had high vaccination coverage of routine immunization, with a national vaccination rate ranging from 90.9–99.3% [57]. In our study, adults visited the medical center for vaccination without the additional worries of financial burden or medical accessibility after the COVID-19 pandemic. As a result, while the world showed a decreased rate of vaccine administration due to the lockdown, inconvenience in visiting doctors, and shortage of medical services, we found an apparent rising rate of pneumococcal and influenza vaccines per contra.

We discovered that public attitudes towards the COVID-19 pandemic influenced vaccine hesitancy not only for COVID-19 vaccines but also for pneumococcal and influenza vaccines. Vaccine hesitancy is a phenomenon in which, despite the abundance of evidence supporting vaccination and immunization worldwide, people still refuse to get vaccinated or delay acceptance. In our opinion, the COVID-19 outbreak is a novel event, and Taiwanese citizens are not confident in COVID-19 vaccines. Since the pandemic was under control in Taiwan before May 2021, the vaccine acceptance rate was much lower than in neighboring countries at the beginning of vaccination campaign [58]. After the Taiwanese government issued a Level 3 pandemic alert on 15 May 2021, and enforced home quarantine, social distancing, school lockdowns, and prohibited public gatherings, fewer than 1% of the population was vaccinated [59]. This was not due to vaccine hesitancy but rather insufficiency for most residents, which led to widespread panic [60]. This unstable attitude drew our attention to health literacy in Taiwan. One study estimated that thirty percent of adults in Taiwan lack adequate or marginal health literacy [61], and another report indicated that the general health literacy of the Taiwanese population was 34.4 ± 6.6 out of 50, as measured using the European Health Literacy Survey Questionnaire (HLS-EU-Q) [62]. To better define and understand the primary determinants of vaccine uptake, vaccination literacy examines health literacy from the perspective of vaccine attitudes and hesitancy. We believe our study demonstrates that, although Taiwanese citizens initially lacked confidence in and interest in the new COVID-19 vaccines, most residents were still aware of the importance of immunization and were willing to seek medical advice and receive other kinds of vaccines to prevent severe illness in the absence of COVID-19 vaccines.

### 4.5. Strengths and Limitations

The importance of pneumococcal and influenza vaccines during the COVID-19 pandemic has been emphasized repeatedly, and one of the guiding principles of the WHO European Region for immunization programs during the COVID-19 pandemic has strongly asserted that prioritizing pneumococcal and seasonal influenza vaccines to protect vulnerable populations is imperative [63]. To our knowledge, this is the first report comparing the administration times of influenza and pneumococcal vaccines (PCV13 and PPSV23) before and after the COVID-19 outbreak in medical centers in Taiwan. Public health officials can use this information to estimate trends in public vaccination attention to protect the population in a given year. Considering that multiple respiratory coinfections have been frequently reported in previous studies, we conclude that public health programs must collaborate to enhance campaigns to prevent pneumococcal and influenza infections during the COVID-19 pandemic. Special attention should be paid to patients who are more likely to have a negative prognosis after contracting COVID-19.

This study has several limitations. First, it only included a partial sample from Taiwan. Although CGMH institute are one of the largest medical centers in Taiwan (Linkou and Kaohsiung branches ranked in the Top 3 of total profits in the Taiwan health insurance database for at least 3 years since 2018) and include data from southern to northern Taiwan, we still lack results from other hospitals and even from different countries. Therefore, the results may be biased and not necessarily representative. Despite the relatively small sample sizes, the researchers attempted to capture a diverse population with multiple chronic diseases, timings, sexes, and ages from a finite sample to represent vaccination behaviors after COVID-19. Second, since we collected data from the Chang Gung Research Database for adults who used diagnostic code “Z23-encounter for immunization,” and aimed for those who were prescribed with influenza, PSV13, or PPSV23 vaccines, there is a possibility that doctors did not order a diagnostic code at that time, which could lead to several patients being omitted. Third, COVID-19 was first detected globally in December 2019 and the first case in Taiwan was detected in January 2020 [17,18,19]. We defined hospital cases from January 2018 to December 2019 as “before COVID-19 outbreak,” and hospital cases from January 2020 to December 2021 as “after COVID-19 outbreak”. However, Taiwan has faced its first major COVID-19 surge in May 2021, and a great number of Taiwanese people perceived the importance of getting a COVID-19 vaccine and other vaccinations for protection [64]. Hence, a major surge may draw public attention to vaccinations at different times. In our study, we covered the period from the first diagnosed case to the outbreak in May 2021; however, the long-term impact remains unclear, and further follow-ups on vaccine coverage and nationwide surveillance will be valuable.

## 5. Conclusions

The present study revealed an increased willingness to receive both influenza and pneumococcal vaccinations among women, adults without underlying disease, and younger adults since the COVID-19 outbreak. The COVID-19 pandemic may have increased awareness of the importance of vaccination in Taiwan. Future studies are required to determine the response of influenza, pneumococcal, and COVID-19 vaccines. Although COVID-19 is no longer strictly monitored due to its global vaccination coverage, this study will enable medical professionals to further monitor and predict the future occurrence of the disease, helping to prepare adequate vaccines and raise medical awareness among the population.

## Figures and Tables

**Figure 1 vaccines-11-00986-f001:**
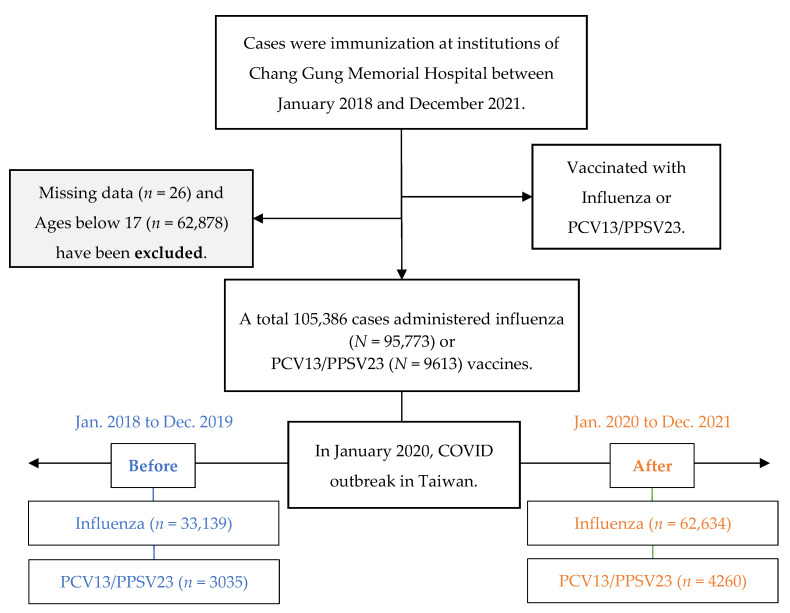
Flowchart of inclusive criteria for the study. Abbreviations: pneumococcal 13-valent conjugate vaccine (PCV13) and pneumococcal polysaccharide vaccine (PPSV23).

**Figure 2 vaccines-11-00986-f002:**
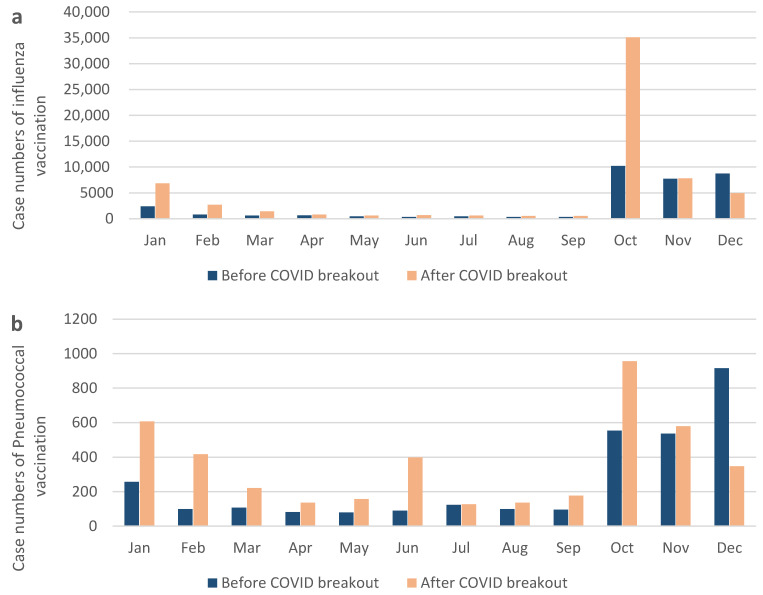
Number of cases of (**a**) influenza vaccination and (**b**) pneumococcal vaccination before and after COVID outbreak.

**Figure 3 vaccines-11-00986-f003:**
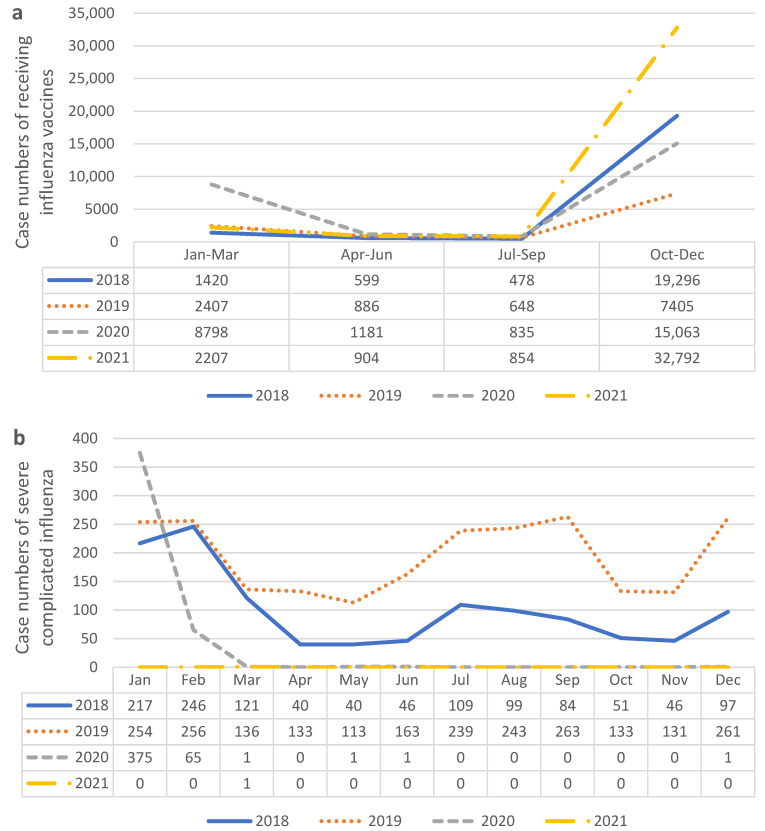
(**a**) Numbers of cases receiving influenza vaccines (**b**) Numbers of cases with severe complicated influenza in Taiwan. Data from the Taiwan Centers for Disease Control.

**Table 1 vaccines-11-00986-t001:** Characteristics of cases injected with influenza vaccine before and after COVID-19 outbreaks. (*n* = 95,773).

	Before Outbreak (*n* = 33,139)	After Outbreak (*n* = 62,634)	*p*-Value
Age	66 ± 13.8	54 ± 18.3	<0.001
Male sex	16,471 (49.7%)	25,319 (40.4%)	<0.001
Location			
Kaohsiung institution	24,170 (72.9%)	36,734 (58.6%)	<0.001
Chiayi institution	706 (2.1%)	3465 (5.5%)	
Linkou institution	5956 (18%)	18,079 (28.9%)	
Keelung institution	2307 (7%)	4356 (7%)	
Without underlying diseases	9298 (28.1%)	31,554 (50.4%)	<0.001
Underlying diseases			
Hypertension	16,725 (50.5%)	21,143 (33.8%)	<0.001
Diabetes mellitus	11,485 (34.7%)	14,342 (22.9%)	<0.001
Liver cirrhosis	1177 (3.6%)	1266 (2%)	<0.001
End stage renal disease	971 (2.9%)	1472 (2.4%)	<0.001
Coronary artery disease	4515 (13.6%)	5191 (8.3%)	<0.001
Heart failure	1989 (6%)	2140 (3.4%)	<0.001
Cerebrovascular accident	5135 (15.5%)	6882 (11%)	<0.001
Malignancy	5770 (17.4%)	7511 (12%)	<0.001

Data were presented as a number (percentage) and mean ± standard deviation.

**Table 2 vaccines-11-00986-t002:** Characteristics of cases injected with pneumococcal vaccine before and after COVID-19 outbreaks. (*n* = 7295).

	Before Outbreak (*n* = 3035)	After Outbreak (*n* = 4260)	*p*-Value
Age	72 ± 12.1	68 ± 14.5	<0.001
Male sex	1508 (49.7%)	2011 (47.2%)	0.037
Location			
Kaohsiung institution	1953 (64.3%)	2617 (61.4%)	0.240
Chiayi institution	125 (4.1%)	193 (4.5%)	
Linkou institution	670 (22.1%)	966 (22.7%)	
Keelung institution	287 (9.5%)	484 (11.4%)	
Vaccine type			
PCV13	1366 (45.0%)	2620 (61.5%)	<0.001
PPSV23	1669 (55.0%)	1640 (38.5%)	
Without underlying diseases	807 (26.6%)	1320 (31.0%)	<0.001
Underlying diseases			
Hypertension	1675 (55.2%)	2157 (50.6%)	<0.001
Diabetes mellitus	1181 (38.9%)	1512 (35.5%)	0.003
Liver cirrhosis	95 (3.1%)	111 (2.6%)	0.183
End stage renal disease	91 (3%)	143 (3.4%)	0.392
Coronary artery disease	379 (12.5%)	534 (12.5%)	0.952
Heart failure	199 (6.6%)	263 (6.2%)	0.508
Cerebrovascular accident	499 (16.4%)	698 (16.4%)	0.949
Malignancy	531 (17.5%)	682 (16%)	0.093

Data were presented as a number (percentage) and mean ± standard deviation. Pneumococcal 13-valent conjugate vaccine (PCV13) and pneumococcal polysaccharide vaccine (PPSV23).

## Data Availability

Data is unavailable due to privacy or ethical restrictions.

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
