# Peer review of "Impact of COVID-19 Outbreak on Influenza and Pneumococcal Vaccination Uptake: A Multi-Center Retrospective Study"

_vaccines, 2023, doi:10.3390/vaccines11050986_

Round 1

Reviewer 1 Report

The authors conducted a large-scale study taking into account the period of incidence. They referred in their results to predecessors who had obtained similar results. In my opinion, the article can be accepted for publication after minor revisions according to the comments below.

1.Line 28 – there is a typo – “?” instead of “.”.

2.Section 2.1. The source and producer of vaccine should be indicated here.

3.Lines 134-136. Looking at the text, the authors first referred to Fig. 3a. However, first of all, all the figures that make up Fig. 3, and then only from the next section, the citation of other parts of figure 3 begins. It is necessary to first refer to fig. 3 as a whole, then into its separate parts.

Please check typos and grammar before finalizing paper

Author Response

Point 1: Line 28 – there is a typo – “?” instead of “.”.

Response 1: We have corrected the mistake. Thank you.

Point 2: Section 2.1. The source and producer of vaccine should be indicated here.

Response 2:

We appreciate the reviewer for your time and expertise in helping us improve our manuscript.

In Chang Gung Memorial hospital, we provided PPSV23(PNEUMOVAX®) from MSD which is government-funded in Taiwan for adults over 75 years old. PCV13(Prevenar 13®) from Wyeth at individuals’ expense if they chose to receive PCV13 instead of PPSV23. Influenza vaccine included Vaxigrip Tetra®, Fluarix Tetra®, AdimFlu-S(QIS)® and FLUCELVAX QUAD®. These vaccines provided randomly to public in Taiwan according to purchase amount by government since October every year. (line88-95)

Point 3: Lines 134-136. Looking at the text, the authors first referred to Fig. 3a. However, first of all, all the figures that make up Fig. 3, and then only from the next section, the citation of other parts of figure 3 begins. It is necessary to first refer to fig. 3 as a whole, then into its separate parts.

Response 3: We are grateful for your suggestion. We have adjusted the content according to your suggestion. (line 158-162)

Reviewer 2 Report

Abstract: there are some minor issues, please revise it to improve readability (e.g. this sounds a bit odd In addition, there was an increased willingness to receive both influenza and pneumococcal vaccinations among women? adults without underlying disease and younger adults). 

Introduction 

- line 36: "Annual influenza vaccination is recommended for all adults without contraindications". This is not entirely true for all countries worldwide, as in many countries (e.g. in some EU countries) the word "recommended" means than that the authorities will offer the vaccination for free. Therefore, for different reasons, it is often "recommended" to >65 years or HCPs (as an example) and "suggested" to healthy adult individuals, to make a distinction. Since what you write is true for the US (1), I suggest rephrasing and stick to some specific situations (e.g. In the US, Routine annual influenza vaccination is recommended for all persons aged ≥6 months who do not have contraindications). 

- line 55: despite being true that influenza vaccination coverage rates increased after the first year of COVID-19 (relevant to mention that not only in the UK (2)), I would be careful in reporting ecological studies that link higher influenza vaccination rates and better COVID-19 outcomes, as at the moment (to the best of my knowledge) there is no strong biological reason to affirm this, even if in recent times a number of systematic reviews tried to assess it (3-6). 

- line 74 (whole paragraph): again, I find this a hazardous speculation, as there is no (at the moment) biological reason and high-quality, strong studies to affirm that pneumococcal vaccines can prevent COVID-19 infections or poor outcomes. Since this is not the focus of your study and my feeling is that this paragraph could be removed without harm from the introduction, I suggest removing it (lines 74-81) and directly go to the aim of the study. 

Methods: the section is overall well-written, but I suggest expanding it a bit and provide more information (e.g. state also in the methods that is a retrospective study; explain how the participants were selected (e.g. all consecutive patients attended at the hospital?); explain what kind of informed consent they sign to be recruited for a retrospective study). 

Moreover, you state that "Individuals with multiple visits to CGMH institutions were counted as multiple cases". This sounds fine if you did this for patients visiting in different seasons (which means different vaccine and different pattern of circulation), but how did you handle data from patients coming twice or three times during the same year/season? I suggest giving explanations and details on all this issues to give a reader a better understanding of your study. 

Results: please make sure all numbers have "," as separator for thousands and "." for decimals. 

- In figure 1. (colored parts, "before" and "after", please specify the two time range considered). 

- If your work included all patients that got vaccinated against influenza and pneumococcus in that time range, it is important to also report the total number of patients attended in the same institutions in the same period. To have a chance to calculate a % of vaccination would be more informative than just reporting raw numbers (also important, but not that informative without a proper denominator). Also, if your work DOES not include all patients vaccinated against flu and pneumo, please specify how many. 

- How do you explain the statistical significances that you report in table 1? They basically say that the two populations considered were totally different in terms of location and underlying diseases pre and post covid-19. How do you explain such a different distribution in patients visiting your institutions? 

- Discussion: I do not understand the first two paragraphs of the discussion. Your study aims investigating whether the COVID-19 pandemic has changed the public behavior towards influenza vaccine and pneumococcal vaccine in Taiwan (a very relevant aim!), not the impact of the vaccinations on COVID-19. I suggest rewriting it consistently with the aim of the study, considering you are not providing data to argue whether the vaccinations have a good impact on COVID-19 or not. 

Paragraph 4.4 is interesting and goes to the point: how do you explain the high vaccination coverage rates in Taiwan (before and after COVID?). Do you think that high health/vaccine literacy (7) had a role on it? Specifically, since you correctly mentioned that  "Taiwan’s health insurance program is distinctive [...] resulting in good accessibility, low cost, comprehensive coverage and short waiting times" it would be interesting to have your point of view on whether the institutions included in your study could be considered health-literate (8) and whether this had also a role in making sure people had access to vaccination. 

Some papers for consideration:

1. https://www.cdc.gov/flu/professionals/acip/summary/summary-recommendations.htm

2. https://pubmed.ncbi.nlm.nih.gov/36553093/

3. https://pubmed.ncbi.nlm.nih.gov/34065294/

4. https://pubmed.ncbi.nlm.nih.gov/36378238/

5. https://pubmed.ncbi.nlm.nih.gov/29048987/ 

6. https://pubmed.ncbi.nlm.nih.gov/32268620/

Minor editing of English language required

Author Response

Point 1:

Abstract: there are some minor issues, please revise it to improve readability (e.g. this sounds a bit odd In addition, there was an increased willingness to receive both influenza and pneumococcal vaccinations among women? adults without underlying disease and younger adults).

Response 1: We have corrected the mistake. Thank you.

Introduction

Point 2:

- line 36: "Annual influenza vaccination is recommended for all adults without contraindications". This is not entirely true for all countries worldwide, as in many countries (e.g. in some EU countries) the word "recommended" means than that the authorities will offer the vaccination for free. Therefore, for different reasons, it is often "recommended" to >65 years or HCPs (as an example) and "suggested" to healthy adult individuals, to make a distinction. Since what you write is true for the US (1), I suggest rephrasing and stick to some specific situations (e.g. In the US, Routine annual influenza vaccination is recommended for all persons aged ≥6 months who do not have contraindications).

Response 2: We appreciate the reviewer for your time and expertise in helping us improve our manuscript. We have corrected the phrase that I use to specific the country. (line36-37)

Point 3:

- line 55: despite being true that influenza vaccination coverage rates increased after the first year of COVID-19 (relevant to mention that not only in the UK (2)), I would be careful in reporting ecological studies that link higher influenza vaccination rates and better COVID-19 outcomes, as at the moment (to the best of my knowledge) there is no strong biological reason to affirm this, even if in recent times a number of systematic reviews tried to assess it (3-6).

Response 3: Thank you for your valuable suggestion. From line 55-57, I tend to bring up public might not want to seek medical care (including getting annual influenza vaccine) after COVID-19 pandemic. However, on the other hand, I mentioned several reports which might affect public to NOT getting COVID-19 vaccine, and that’s why people may choose vaccines that protect against other infectious respiratory diseases instead of COVID-19 vaccination for compensation. (line57-60). In this paragraph, we try to emphasize the controversial opinions whether influenza vaccine brings better COVID-19 outcomes. (line62-65)

Point 4:

- line 74 (whole paragraph): again, I find this a hazardous speculation, as there is no (at the moment) biological reason and high-quality, strong studies to affirm that pneumococcal vaccines can prevent COVID-19 infections or poor outcomes. Since this is not the focus of your study and my feeling is that this paragraph could be removed without harm from the introduction, I suggest removing it (lines 74-81) and directly go to the aim of the study.

Response 4: We appreciate the reviewer for your time and expertise in helping us improve our manuscript. We deleted most of the content in this paragraph as your suggestions. We try to bring up some various ideas about the relationship between influenza and pneumococcal vaccines in the first place, but we understand and agree with you that focusing on speculation may complicated our expression, thank you.

Point 5:

Methods: the section is overall well-written, but I suggest expanding it a bit and provide more information (e.g. state also in the methods that is a retrospective study; explain how the participants were selected (e.g. all consecutive patients attended at the hospital?); explain what kind of informed consent they sign to be recruited for a retrospective study).

Moreover, you state that "Individuals with multiple visits to CGMH institutions were counted as multiple cases". This sounds fine if you did this for patients visiting in different seasons (which means different vaccine and different pattern of circulation), but how did you handle data from patients coming twice or three times during the same year/season? I suggest giving explanations and details on all this issues to give a reader a better understanding of your study.

Response 5: We have added some explanations of how it works in Taiwan to get immunization in outpatient system which affect our data collection, hope to specify the methods. (line99-103, 109-118)

Point 6:

Results: please make sure all numbers have "," as separator for thousands and "." for decimals.

Response 6: We have corrected all the numbers. Thank you.

Point 7:

- In figure 1. (colored parts, "before" and "after", please specify the two time range considered).

Response 7: Thank you for your suggestion. We had specified the time range on the figure 1.

Point 8:

- If your work included all patients that got vaccinated against influenza and pneumococcus in that time range, it is important to also report the total number of patients attended in the same institutions in the same period. To have a chance to calculate a % of vaccination would be more informative than just reporting raw numbers (also important, but not that informative without a proper denominator). Also, if your work DOES not include all patients vaccinated against flu and pneumo, please specify how many.

Response 8: Thank you for the suggestion. We have added the results of total outpatient visit in 2018-2021, and describe the difference in percentage (line 132-142). Also, line 311-318 narrate the discussion of this finding as your suggestions.

Point 9:

- How do you explain the statistical significances that you report in table 1? They basically say that the two populations considered were totally different in terms of location and underlying diseases pre and post covid-19. How do you explain such a different distribution in patients visiting your institutions?

Response 9: We are grateful for your suggestion. We mentioned the differences of location and hospital scale, in order to explain the result of cases among different branches in our institute. (line151-155)

Point 10:

- Discussion: I do not understand the first two paragraphs of the discussion. Your study aims investigating whether the COVID-19 pandemic has changed the public behavior towards influenza vaccine and pneumococcal vaccine in Taiwan (a very relevant aim!), not the impact of the vaccinations on COVID-19. I suggest rewriting it consistently with the aim of the study, considering you are not providing data to argue whether the vaccinations have a good impact on COVID-19 or not.

Response 10: We appreciate your opinions and expertise in helping us improve our research. We have rewrote the first two paragraphs and mostly aim for the view of why COVID-19 outbreak lead to more attention of pneumococcal and influenza vaccines. (Discussion 4.1-4.2)

Point 11:

Paragraph 4.4 is interesting and goes to the point: how do you explain the high vaccination coverage rates in Taiwan (before and after COVID?). Do you think that high health/vaccine literacy (7) had a role on it? Specifically, since you correctly mentioned that "Taiwan’s health insurance program is distinctive [...] resulting in good accessibility, low cost, comprehensive coverage and short waiting times" it would be interesting to have your point of view on whether the institutions included in your study could be considered health-literate (8) and whether this had also a role in making sure people had access to vaccination.

Response 11: Thank you for your valuable questions and suggestions. We have added some descriptions of vaccine hesitancy and health literate affecting Taiwanese rushed to get pneumococcal and influenza vaccines. This is truly an important point of view, however, studies of health literate in Taiwan are scarce, and it narrows our perspective to hypothesis mainly. (line 328-350)

Round 2

Reviewer 2 Report

Dear authors, 

Thank you for your answers and for addressing all the comments. 

English is pretty much fine